# AR-Supported Supervision of Conditional Autonomous Robots: Considerations for Pedicle Screw Placement in the Future

**DOI:** 10.3390/jimaging8100255

**Published:** 2022-09-21

**Authors:** Josefine Schreiter, Danny Schott, Lovis Schwenderling, Christian Hansen, Florian Heinrich, Fabian Joeres

**Affiliations:** 1Faculty of Computer Science & Research Campus STIMULATE, University of Magdeburg, 39106 Magdeburg, Germany; 2Innovation Center Computer-Assisted Surgery (ICCAS), Faculty of Medicine, Leipzig University, 04103 Leipzig, Germany

**Keywords:** augmented reality, workflow analysis, pedicle screw placement, human–robot interaction, conditional autonomous robots

## Abstract

Robotic assistance is applied in orthopedic interventions for pedicle screw placement (PSP). While current robots do not act autonomously, they are expected to have higher autonomy under surgeon supervision in the mid-term. Augmented reality (AR) is promising to support this supervision and to enable human–robot interaction (HRI). To outline a futuristic scenario for robotic PSP, the current workflow was analyzed through literature review and expert discussion. Based on this, a hypothetical workflow of the intervention was developed, which additionally contains the analysis of the necessary information exchange between human and robot. A video see-through AR prototype was designed and implemented. A robotic arm with an orthopedic drill mock-up simulated the robotic assistance. The AR prototype included a user interface to enable HRI. The interface provides data to facilitate understanding of the robot’s ”intentions”, e.g., patient-specific CT images, the current workflow phase, or the next planned robot motion. Two-dimensional and three-dimensional visualization illustrated patient-specific medical data and the drilling process. The findings of this work contribute a valuable approach in terms of addressing future clinical needs and highlighting the importance of AR support for HRI.

## 1. Introduction

For many years, medical robotics has been a hot research topic [1]. In the coming years, it can be assumed that more and more robots will be involved in clinical routines. In contrast with today’s robotic systems, which have a low degree of autonomy due to legal and ethical barriers, as well as a lack of integration into clinical workflows, the autonomy of these systems is expected to increase in the future [2]. However, assuming that technological advances will outpace regulatory and ethical frameworks, autonomous robots will need to be supervised by humans in the mid-term. For these supervised robotic procedures, it is crucial to establish effective human–robot interaction (HRI) in order to enable information exchange between the two agents. Augmented reality (AR) could be the way to enable this interaction, allowing the user to instruct the robot and, in turn, allowing the robot to communicate information about its state and actions to the human. Furthermore, additional information, such as medical data superimposed to the surgical site, could improve the surgeon’s situational awareness.

This work provides insights into a possible futuristic scenario of a robotic assistant that enables HRI with the help of AR performing an orthopedic procedure for PSP under human supervision.

### 1.1. Clinical Background

Pedicle screw placement (PSP) is a common technique in orthopedic surgery and an important step for the purpose of spinal fusion to connect two or more vertebrae [3]. This fusion leads to support of the vertebrae and stabilization of the spine, eliminating instability of the treated segment [4]. The technique remains one of the most fundamental procedures in treating lumbar diseases if alternative treatment methods, such as movement-preserving procedures, do not lead to any improvement. In order to allow stability between the vertebrae, screws are inserted into the vertebral bodies, which are connected to each other by rods or metal plates [5]. Two screws are inserted into the pedicles for each vertebra, with heads in posterior orientation (see Figure 1). The screws serve as immovable anchor points, whereas the rod connects the screws, causing fixation of the spinal segment.

The distinct anatomy of the thoracolumbar vertebrae, the deep location within the human body as well as adjacent risk structures, including the spinal cord, nerve roots and major blood vessels, pose a great challenge in terms of ensuring accurate screw placement without injuring these structures [6].

There are various approaches for PSP in the lumbar spine. Open procedures require wide posterior exposure, including separation of the erector spinae muscle and soft-tissue disruption of a portion of the facet joint [5]. In contrast with open procedures, minimally invasive screw placement uses tubular retractors to reach the pedicles. Screws are inserted via these retractors with the use of a small guidewire [5]. Due to limited view and restricted tactile feedback in minimally invasive surgeries, intraoperative imaging is of central importance. Percutaneous insertion of pedicle screws is a safe and reliable technique, reducing the postoperative patient care time and complication rate [7]. On the contrary, open PSP procedures yield risks such as surrounding tissue injury and complications, such as neurologic and vascular damage, and reduced stability of the treated segment [6].

Indications for robot-assisted interventions are, according to Tian et al. [6], a large part of frequently occurring thoracic and lumbar conditions. Robotic systems involved during these interventions are clinically used with or without navigation techniques [4]. In the following section, existing robotic systems are introduced.

### 1.2. Related Work

Our project addresses a clinical scenario from the mid-term future, rather than clinical challenges in today’s operating rooms. Therefore, little current research exists that covers the needs arising in this scenario. However, current research addresses some aspects that we are already incorporating into our work. These aspects include recent advances in robot-assisted orthopedic surgery and AR visualization, which are reported below.

#### 1.2.1. Robot-Assisted Orthopedic Surgery

In recent decades, robotic systems have been introduced for the placement of pedicle screws within the thoracolumbar spine [8]. In comparison with free-hand techniques, where screws are placed by surgeons according to anatomic landmarks with the help of image navigation, robot-assisted approaches achieve better clinical outcomes by eliminating human hand tremors. Thus, accuracy and precision of the screw placement are increased with reduced intervention time and radiation exposure [6,8]. In the work of Kantelhardt et al. [9], the placement accuracy of pedicle screws was found to be higher for robot-assisted placement (94.5%) compared to open, conventional procedures (91.4%), with no differences found between open and minimally invasive robotic placements.

The current FDA-approved commercial systems offering the latest technologies for spine surgery are Mazor X^TM^ Stealth Edition (Medtronic plc, Dublin, Ireland), ROSA^®^ ONE Spine (Zimmer Biomet Holdings Inc., Warsaw, IN, USA) and Excelsius GPS^TM^ (Globus Medical Inc., Audubon, PA, USA). These robotic systems are operated in a similar manner. The workflow resembles the following steps (according to Jiang et al. [8]):Preoperative Planning: preoperative imaging and planning of the screw trajectory with the help of the system software.Mounting: positioning of the robot on the spine of the patient.Registration: matching of preoperative images with intraoperative anatomy.Operation: robotic guidance to place screws.

Yang et al. [2] investigated the autonomy levels of medical robotic systems. The levels were defined depending on the autonomy with which a system independently makes and realizes decisions. Level zero represents no robot autonomy, as can be found in tele-operated robots. The first level introduces robotic assistance, which is provided by mechanical guidance while the robot is continuously controlled by the operator. The second level includes task autonomous robots being able to perform specific tasks initiated by human hands, e.g., suturing. Here, the operator has discrete rather than continuous control of the system. Level three describes conditional autonomy, in which surgical strategies are generated and proposed by the system and thereafter selected by the operator. High autonomy on level four means that the robotic system makes medical decisions and performs them by itself while being supervised by a surgeon. Finally, fully autonomous robots are not supervised at all.

Current commercial systems refer to low autonomy levels of zero and one. The above described systems provide guidance along a human-planned trajectory, while drilling and screw placement are performed manually by surgeons.

Recent advances show the potential of higher autonomy levels. Smith et al. [10] proposed an image-guided, supervisory controlled autonomous spine surgery system that automates the process of bone preparation and polyaxial screw insertion. Siemionow et al. [11] developed a robotic system to autonomously plan the placement of lumbar pedicle screws based on machine learning techniques to identify parameters of the screws. A conjunction of those prototypes may very well result in a fully autonomous robotic system. In the mid-term, however, we anticipate robotic systems with conditional autonomy, referring to level three of the described scale. Systems of these autonomy levels are currently not commercially available, but are a subject of recent research.

#### 1.2.2. Augmented Reality Visualization

AR visualization provided to surgeons during robotic interventions is an intriguing field of research. A general review about AR for robotics was conducted by Makhataeva et al. [12]. Here, several application areas, use cases and current research directions are summarized. Many works focus on the superimposition of virtual visual cues, such as human anatomy, to improve the user’s perception [13] and the display of robotic tool trajectory [13,14]. In a work of Zhou et al. [15], a robotic system based on AR was introduced that enables visual instructions for preoperative planning and visual force feedback. Liu et al. [16] report an AR system visualizing a decision tree of their robotic system for opening bottles, so that decisions made by artificial intelligence can be better understood. In addition, their work also includes a visualization of the end-effector force, which may be a meaningful addition for our prototype. Iqbal et al. [17] developed a mixed-reality workflow of a robot-assisted orthopedic intervention with the help of AR. Holographic information, including 3D medical image data and an immersive user interface, were overlayed on the real surgical side. Results show less user interaction with data on an external navigation screen. Changes in the surgeon’s view direction and the mental load were reduced, while the standard surgical workflow was not significantly altered.

The usage of AR for robot-assisted surgeries is low compared to AR implementation for free-hand techniques. In the case of PSP, several articles describe solutions and their evaluation. Dennler et al. [18] and Elmi-Terander et al. [19] display anatomical information and the drilling trajectory, and thus improve the drilling precision. Elmi-Terander et al. [19] additionally show the planned and current surgery plan to the supervisor.

For anatomical structures, correct spatial perception should be supported, e.g., using occlusion masks or virtual windows [20] and context visualization [21]. Different examples on how to display linear trajectories in surgery were reported by Schütz et al. [22] and Heinrich et al. [23]. In addition, Khlebnikov et al. [24] introduced the representation of linear paths by means of crepuscular rays. Although not intended by the authors, this more scattered visualization could be an interesting method to visualize uncertainty resulting from registration inaccuracy.

### 1.3. Contribution

We envision a futuristic scenario taking place a few years from now, when legal matters around artificial intelligence have been resolved. This project explores a potential surgical workflow developed in collaboration with medical and robotic experts based on the current clinical procedure of the selected intervention. We developed a prototypical AR assistant for supervised robotic PSP procedures. As such, the project takes a new look at how to provide surgeons with meaningful visualization in HRI and contributes in the section of AR support for surgical staff in a devised clinical setting of the future.

## 2. Materials and Methods

The methodology of this project followed the principles of *Design Science Research* according to Hevner et al. [25]. This approach incorporates the iteration of three cycles to develop a solution to a research problem and the subsequent analysis of its performance. The cycles complement each other and include the definition of the context in which the problem occurs (*Relevance Cycle*), analysis of the existing knowledge, theories and related existing solutions of the problem (*Rigor Cycle*), and design and implementation of the solution as well as its evaluation (*Design Cycle*). The methodology of our project comprised the *Relevance Cycle* by defining the application context, determining the requirements for the solution to be developed and defining the criteria that characterize the solution. The *Rigor Cycle* involved the conduction of a literature review of related work and the *Design Cycle* comprised the design of the AR concept, the selection of the used hardware and the practical implementation resulting in a prototype. The subsequent evaluation as part of the *Design Cycle* was omitted in our project and is considered a future project step. Moreover, the methodology followed a user-centered approach by involving clinical and robotic experts in the process of the requirement analysis. In applying the above methodology, we followed the methods shown in Figure 2, which are explained in the following subsections.

### 2.1. Current Workflow

The project does not address present clinical needs, but looks at future challenges in an automated surgical scenario. In order to build this scenario, the current robot-assisted PSP procedure was first analyzed in detail according to Tian et al. [6]. The workflow was formalized into a task analysis and examined for the clinical goal for each phase. A discussion with a clinical expert was conducted to review the workflow.

### 2.2. Futuristic Workflow

Based on the analysis of the current workflow, the futuristic workflow was elaborated together with a robotics researcher and a senior orthopedic surgeon. Together with the robotics researcher, we defined the robot’s capabilities in the futuristic scenario. Objectives to be achieved within each step of the current robotic-assisted PSP workflow were maintained in the futuristic workflow, as they are considered to be of consistent importance in the mid-term. On the basis of the assumed capabilities of the robotic system, we defined how these clinical objectives of a future PSP procedure would be achieved by allocating tasks to the surgeon, to the robot or to the collaboration between both. A discussion with a senior orthopedic surgeon was performed to evaluate the futuristic workflow.

#### Information Architecture

In addition, the futuristic workflow analysis included the identification of required information to be exchanged between the two agents for each subtask. Data input and output were defined, meaning information that needs to be communicated from the human to the robotic system and vice versa. In the following, two representative steps of the developed futuristic workflow were selected, for which it was considered beneficial to develop AR support.

### 2.3. Prototype

Subsequently, an AR concept was designed for the two selected workflow steps based on the analyzed information exchange. Assumptions were made about possible future hardware. Based on this, appropriate current hardware was selected that most closely matched these assumptions. Thus, this hardware was used to realize the prototype, which consists of a robotic simulation and an AR assistant.

## 3. Results

The outcomes of the project include the analyses of the current and futuristic workflow, as well as the development of a conceptual prototype. The results are explained in the following sections. The resulting prototype is also demonstrated in a Appendix A (https://youtu.be/9yUGJbP7Y9Y (accessed on 12 September 2022)).

### 3.1. Current Workflow

Main workflow phases and their subordinate steps of a current robot-assisted PSP can be seen in Figure 3. The first phase of the intervention aims to prepare the patient and the involved equipment. In the subsequent planning phase, the registration of the robot with the patient and the preoperative data is achieved. Furthermore, the geometry and the equipment of the screw placement are defined. During the execution phase, the robot first positions itself along the path to penetrate the bone. Thereafter, the surgeon manually drills into the bone while the robot provides mechanical guidance. In this manner, the screw is placed in the pedicle. In the final wrap-up phase, the screw placement is verified. Planning of a further screw insertion or fine tuning of the already-placed screws could follow before the intervention is completed.

Each workflow step was analyzed for its level of collaboration between the surgeon and the robot to achieve the defined objective. The collaboration level varies between a human–operator task, a collaborative human–robot task and an autonomous (supervised) robot task. Whereas the surgeon is in charge of completing most of the tasks, three workflow steps are completed in collaboration with the robotic system. In current PSP procedures, solely the task of *robot positioning* according to the drilling path is performed autonomously by the robot under supervision of the surgeon.

### 3.2. Futuristic Workflow

Together with a robotics researcher, the robot’s capabilities in the futuristic scenario were defined. The term *robot* encompasses a system of the mid-term future with the abilities of low-dose CT imaging, image analysis and robotic agency capabilities. The detailed features and related technical implementation of such a system were disregarded, as they are not essential in the scope of the project. Rather, the framework capabilities of the robotic system were defined and a simulation of it was developed.

In contrast with the collaboration level of today’s PSP workflow, higher robot autonomy is assumed in the future, and thus, more tasks were allocated to the robotic system (see Figure 3). The *patient tracker installation* and *sleeve installation* were neglected due to the advanced capabilities of the robot. We consider supervised robot tasks to be autonomously performed by the robot, which may involve manual triggering, confirmation or abortion of the task by the surgeon. These autonomous robot tasks of the futuristic workflow include steps which are currently performed solely by the surgeon or in collaboration with the robot. The workflow steps of *operation planning* and *image verification*, which are nowadays performed by the surgeon, are expected to be completed in the future in collaboration. Four other workflow steps are considered to be performed autonomously by the robot compared to the current workflow.

In the following, more detailed analysis of the futuristic workflow, we focused on the second and third workflow phase, *planning* and *execution* (see Figure 4), since these phases contain the most important steps, and robot autonomy is particularly relevant. Objectives from the current workflow were adopted, and steps as well as subtasks were derived from them. The subtasks were allocated to either one of the agents or to the collaboration of both (see Figure 4A). We categorized subtasks into overarching domains, such as active, displaying and imaging tasks for the robot and active and supervision tasks for the human. If the surgeon only supervises the robot during a subtask (including commands such as triggering, confirmation or abortion), it was categorized an autonomous (supervised) robot task overall.

The discussion with a senior orthopedic surgeon to evaluate the futuristic workflow revealed the importance of an additional workflow step and another subtask, which were then added. In the current workflow, the *incision* to the vertebra is performed independently by the human during the preparation phase. We assumed that in the future, the robot will be able to perform this task autonomously based on the planning data generated collaboratively in the previous workflow step. In addition, the assessment of bone cementing and cement insertion in case of osteoporotic vertebrae was added as a subtask to the workflow step *screw placement*. We anticipated that, in the future, the necessity of bone cementing will be evaluated by the imaging analysis capabilities of the robotic system based on the acquired image data. Nowadays, the assessment is based on Hounsfield units of interventional CT images [26]. Bone cementing was rated by the surgeon as a potentially dangerous step due to the risk of cement embolism, and was therefore integrated into the workflow.

### 3.3. Information Architecture

The required information exchange between the two agents was analyzed for each subtask (see Figure 4B). Throughout the entire workflow, it is necessary to display certain information, such as the robot state, the used instrument, patient-specific personal data, the current workflow phase, the activity of image acquisition and the radiation load. In addition, the surgeon should have an overview of the workflow with the next and previous steps and should be able to review the progress of the procedure up to the current point in time. The information we considered relevant to be communicated from the system to the human comprises motion prediction, progress visualization and data display, including summaries and assessments with the data that are considered during the decision making. The information that the human inputs to the system, on the other hand, can be generalized to the areas of parameter adjustments and trigger actions, including stop, confirmation, repetition or correction.

We selected two steps of the developed futuristic workflow, where we assumed that the surgeon can particularly benefit from AR: the robot *alignment’s review* before vertebrae penetration and the *initial drilling* step (see Figure 4). These workflow steps were analyzed in depth for the required content to be exchanged from robot to human, and vice versa. The exact content was determined for the different system states and categorized by content type as either *information* or *interaction*. The former includes information displayed by the robot to the user and the latter, as the name suggests, requires user input to the system.

### 3.4. Prototype

The development of the AR robotic prototype included the speculation of possible future hardware, the integration of current hardware that most closely matched these assumptions, the conception and implementation of the AR concept, along with the creation of an interactive GUI, as well as 2D and 3D representations.

#### 3.4.1. Assumptions about Future Hardware

Since the futuristic robotic system we imagine does not yet exist, a prototypical simulation was used. The system hardware was selected according to the defined requirements and availability.

With regard to the AR display, the challenge was to estimate the development of AR devices within the next few years and to find a solution for implementation with today’s technical possibilities. Currently, the HoloLens is considered state of the art and is the hardware most frequently used in orthopedic surgery as an optical see-through head-mounted display (HMD) [27]. The user can perceive the environment independently of the AR device, while the virtual content is displayed on semitransparent displays. The small field of view and insufficient computing power are troublesome [28]. There is also a lack of possibilities to influence the representation of reality and to integrate virtual content realistically into the environment. However, past developments of AR and virtual reality (VR) devices in recent years indicate that many existing limitations will be less relevant in the mid-term [29,30]. For our project, it was important to find a technical solution that would allow the implementation of our visualization as independently as possible from current technical challenges. Therefore, we decided against using current optical see-through HMDs. Video-see-through AR allows a more detailed manipulation of the reality display and, thus, a better integration of the virtual content. The problem here is often latency and quality degradation in the image display, which hinders usage of the visualization and leads to cyber-sickness [31]. While these factors preclude actual clinical use of many video see-through devices, it still results in an opportunity to explore AR, regardless of the limitations associated with using optical see-through displays.

#### 3.4.2. System Setup

The developed prototype encompasses several hardware components (see Figure 5). A Valve Index VR headset (Valve Corporation, Kirkland, WA, USA) was used to display the AR concept. The HMD allowed the overlay of camera footage of the real environment in pass-through mode. A tracking space was set up with three base stations and an HTC Vive tracker (HTC Corporation, Taoyuan, Taiwan) attached to the drill. The Valve Index was further equipped with a Stereo IR 170 Camera Module (Ultraleap, Mountain View, CA, USA) to realize hand tracking. The Mixed-Reality Toolkit Version 2.7.2.0 (Microsoft Corporation, Redmond, WA, USA) was used to incorporate hand and voice interaction and to create the user interface. The Unity Editor Version 2020.3.17 (Unity Technologies, San Francisco, CA, USA) on Windows 10 with Visual Studio and C# (Microsoft Corporation, Redmond, WA, USA) was used to implement the prototype.

The drilling robot was simulated by moving a spiral drill along a motion path. For that, a KUKA lbr iiwa (KUKA AG, Augsburg, Germany) robotic arm was operated using the provided software Sunrise.OS Version 1.16. To simulate the drill, a custom-made mockup was attached to the robot flange. The AR application and the robot were connected via a local WiFi network. Motion requests were received from the AR project and passed to the robot. Information about the robot state and robot motions were passed vice versa. If no robot hardware is available, a virtual simulation can be used as a counterpart to the AR application. In this case, all motion requests are displayed as console output, and the robot state is modified and sent accordingly. To test the prototype, the AR application itself must be started, as well as the robot simulation. The two applications are connected via the mentioned local WiFi network and should run on separate work stations.

#### 3.4.3. AR Concept

The AR prototype consists of three major components: a user interface (see Figure 6) containing a heads-up status bar as well as a dynamic GUI, 2D medical visualization as part of the dynamic GUI (see Figure 6) and 3D representations of patient-specific anatomical data (see Figure 7). In the following sections, reasons for their integration are briefly explained before detailed characteristics of the three components are given in the subsequent subsections.

During the two considered workflow steps, it is necessary for the robotic system to display information, e.g., the automatic accuracy assessment, and for the surgeon to react accordingly. To facilitate this, a user interface is integrated into the AR assistant. Secondly, information which needs to be provided from the system to the surgeon includes patient-specific medical data. These are constantly acquired during the *initial drilling* workflow step and need to be displayed accordingly. Therefore, they are integrated as a central element into the dynamic GUI. Additionally, due to the occluded anatomy, it is beneficial to make patient-specific data visible to the surgeon in the form of 3D anatomical models. These should visualize surgical cues superimposed to the patient’s anatomy, and beyond that, enable interactive manipulation.

#### 3.4.4. User Interface

As identified in the analysis of the information exchange, some information should be constantly available to the surgeon throughout the entire surgery. For this purpose, a heads-up status bar is used. Displayed information includes the current workflow step and workflow progress, the used instrument, the robot state and general patient-specific data (see Figure 6).

Furthermore, the developed AR interface enables the surgeon to review data provided by the system, such as CT images, as well as to make parameter adjustments and collaborative refinements. During the *alignment’s review*, the user can access information about the target vertebra, risk structures and surrounding anatomy, as well as the planned drilling path on basis of the acquired CT images. In the subsequent workflow step of *initial drilling*, a prediction of future steps is displayed, and the current progress of the intervention can be followed. Buttons at the bottom of the GUI enable us to instruct the robot to start the *initial vertebra drilling* (once the review of the proposed drilling is completed), to repeat the trajectory alignment, to adjust parameters and to abort the process. Buttons on the left and right side allow changes of settings within the 2D and 3D medical visualization. The functionalities of these buttons, as well as the functionality of the slider, are therefore explained in the following subsections.

The operation of the interface is realized by two implemented interaction methods. The Ultraleap hand tracking enables the operation of the dynamic GUI by clicking directly on buttons with an extended index finger or with a pointing ray and pinch gestures. Sliders can be moved with a grasping gesture. In addition, the GUI can be repositioned using the same grasping gesture. Alternatively, voice instructions have been integrated to communicate discrete commands to the AR interface. These include commands to start, stop, and continue the robot movement during drilling. The menu can be placed in front of the user via a voice command, and visualization components can be activated and deactivated.

#### 3.4.5. Medical 2D Visualization

Patient-specific CT images are provided from three different anatomical orientations in the center of the user interface (see Figure 6). One image is always displayed as larger, and the two other views are arranged smaller next to it. Buttons on the left side can be used to change windowing parameters, to slice through the image stacks and to circle the views. The central CT images are augmented with additional information that can be shown or hidden on demand using buttons on the right side of the GUI after selecting the button *Layer* (see Figure 6). These include visualization of the drill position, drill path, planning parameters, path deviation, anatomical region of interest, risk structures and anatomical context information. The drill is drawn as a solid rectangular shape with dimensions according to the drill’s diameter and its position in tracking space. The drill path is a stippled line starting at the drill tip, following the drill’s trajectory. Planning parameters are visualized as an opaque outline of the planned drill hole with a transparent filling, as well as a stippled line at the center of that hole. In case of path deviation, a bright, triangular shape between drill and planning parameters is shown to indicate the current misplacement. All anatomical layers are colored segmentations of the corresponding regions. They are drawn with an opaque outline and transparent fillings. Additionally, the user can view an interactive prediction of the subsequent drilling process using a slider that appears on the right side of the GUI after selecting the *Prediction* button. Here, a differently colored drill is visualized beneath the actual drill at the position selected with the slider. At the final prediction stage, a 2D approximation of the final screw position is visualized. Finally, measurements within the CT images can be made by selecting the button located in the lower right corner of the GUI. However, due to limited development time, this functionality, as well as windowing capabilities, were omitted in the prototypical implementation.

CT images are acquired after each drilling step, and the displayed scans are exchanged accordingly. These images are shown in the GUI together with the prediction of the subsequent drilling step. For the demonstrator, simulated CT images were generated from a database of abdominal CT datasets [32]. If the user stops the automatic drilling, which is possible at any time, the current progress can be investigated in more detail. An additional visualization of the drilling accuracy and progress is displayed at this point. For this, we implemented a crosshairs-shaped glyph visualization, as proposed by Heinrich et al. [23] for the navigation of linear instrument insertions. This glyph consists of a small circle indicating orientation accuracy with its position with respect to the center of the crosshairs, and a circular filling to represent drilling depth progress.

#### 3.4.6. Medical 3D Visualization

Three-dimensional visualization of patient-specific data are provided using AR. These models are generated based on segmentations of the acquired CT images and are spatially registered to our phantom and the tracked drill. The registration accuracy is indicated by a grid mesh of the patient (see Figure 7a). The grid was also chosen to enhance the perception of depth for internal structures [20,33]. The physical drill is superimposed with a respective 3D model to enable visibility within the phantom. In addition, the identical information is displayed as shown in the 2D visualization, and all augmentations can be shown or hidden in the *Layer* menu of the GUI (see Figure 6). As such, the drill path is shown as a 3D stippled line, and the planning parameters are visualized by a semi-transparent cylinder with an opaque outline at the intended drill hole position. The cylinder’s diameter and height correspond to that hole. Drilling deviation is also indicated by a bright triangle surface between the drill and the planning visualization. In addition, the 3D visualization provides the interactive prediction of the drilling process, as well. For this purpose, a second drill is rendered showing the predicted drill position according to the slider. Here, a 3D model of the screw is visualized at the end of the prediction. All models are rendered semi-transparently with Fresnel effects and an opaque outline to ensure visibility of internal structures and to preserve shape perception (see Figure 7). A virtual copy of the anatomical 3D model can be created using a voice command or by grabbing the visualization using a pointing ray, and is thereupon positioned in the user’s field of view (see Figure 7b). The model can be scaled and positioned using hand gestures to examine anatomical characteristics in more detail.

## 4. Discussion

We discuss not a current clinical need, but rather one that is likely to arise in the next few years. As current research suggests [2,10,11], robots of the mid-term will be equipped with a higher degree of autonomy and will be able to make decisions independently. Nevertheless, we assume that human supervision will be relevant because of ethical issues, among other things. This supervision is likely to be carried out by means of advanced immersive technologies, such as AR, instead of screen replication. Its benefits, and the technological progress, suggest that AR hardware will be widely used in medical procedures. As other research has shown, AR technology has been successfully integrated into robotic-assisted procedures with no negative impact on the workflow, and a positive influence on the ergonomic conditions for the clinicians [17]. We see the biggest advantages of a mixed-reality environment in the ability to overlay the surgical site with dynamic user interfaces and visual guidance cues, such as spatially aligned 3D patient-specific medical data, preventing the surgeon from having to focus on multiple screens. The extent to which the AR solution we developed affects an orthopedic PSP workflow cannot be examined at this time because we are assuming a hypothetical workflow that does not currently exist. However, recent advances show that current PSP procedure benefits from AR support [18,19]. In addition, research confirms that AR interfaces can be applied to enhance HRI, resulting, for example, in a reduced task completion time [34,35].

In comparison to current FDA-approved commercial systems, our system incorporates a robot of conditional autonomy offering surgical strategies and performing them autonomously under supervision throughout the entire workflow. Current integrated software provides monitor-based information, allowing collaborative planning and guidance for screw placement. Future approaches regarding commercial systems indicate the potential of AR to provide visual data on the surgical field rather than integrating a display, eliminating the need to shift attention to the display [8]. Nevertheless, a comparison of the FDA-approved robots to our system has little significance, since we are addressing a futuristic scenario which does not exist in this form today.

Even though AR technology poses great benefits for intraoperative guidance and improved HRI, some limitations need to be considered. Realistic representation of virtual content into the environment is challenging and can lead to perceptual conflicts [36] and inattentional blindness [37]. This includes the disturbed perception between the real 3D world and stereoscopic virtual representations, and thus, the danger of overlooking virtual objects in mixed reality. However, it is expected that, as technology advances, mixed-reality technologies will improve and compensate for the major drawbacks of current hardware.

The results of our work are based on various speculative assumptions about surgical robots of the mid-term future. This, and the hypothetical surgical scenario, rather than a classic approach in system development, make it difficult to conduct a valid user evaluation of the prototype or of the AR concept at this stage. Thus, it is conceivable to test the performance and the usability of the developed AR prototype in a future project step. Moreover, it would be beneficial to involve more experts in the further development process in order to incorporate manifold opinions of potential end users. Another limitation of our work is that no detailed descriptions about workflow steps are given, such as the registration process or how access to the vertebra is achieved. This is due to the fact that, up to the present state of our work, we focused on defining the broad framework capabilities of the robotic system and deriving the workflow from it by outlining major steps rather than close examination of all incorporated procedures. Nevertheless, as a future step, unresolved details regarding the robotic system and the workflow will be defined more precisely.

The primary work of this project included the conception and development of the prototype based on a user-centered approach. Future work will focus on the performance and usability evaluation of the AR prototype, as well as extension of the prototype for further workflow steps. It is envisioned to adopt the methodology of the workflow analysis with elaboration of the information exchange, as well as the development of a mixed-reality interaction concept for other current clinically relevant workflows.

## 5. Conclusions

Our project investigated a futuristic scenario of an orthopedic surgery involving HRI. Since the developed prototype does not address a current clinical need, a hypothetical workflow was elaborated based on the current PSP workflow in collaboration with robotic and medical experts. The user-centered investigation revealed a surgical workflow, from which the required information exchange between the robot and the surgeon was derived. The developed conceptual prototype consists of a simulated robotic system and an AR environment that allows the surgeon to supervise the conditional autonomous robot during PSP.

Overall, we can see the project’s contribution in starting the discussion on how the community can address a clinical need that is likely to arise in the future. Furthermore, it highlights the importance of mixed-reality environments for HRI.

## Figures and Tables

**Figure 1 jimaging-08-00255-f001:**
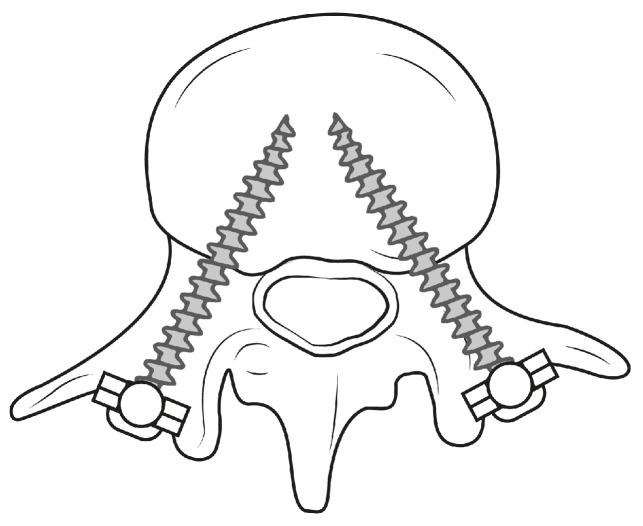
Schematic illustration demonstrating PSP within a lumbar vertebra.

**Figure 2 jimaging-08-00255-f002:**
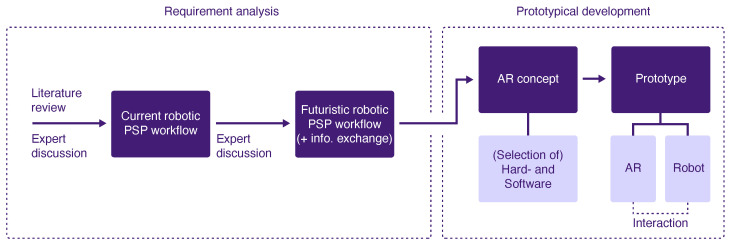
Methodical approach of the project including the analysis of requirements and prototype development.

**Figure 3 jimaging-08-00255-f003:**
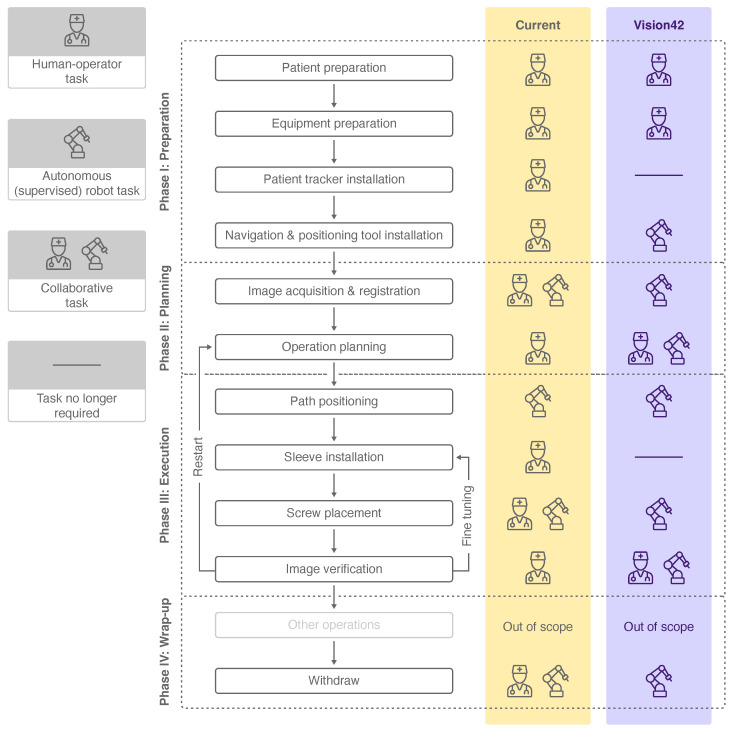
Workflow phases, subordinate steps and human-robot collaboration levels of a current PSP procedure [6] (yellow) and a futuristic PSP procedure (violet). The icons used are explained in the legend on the left.

**Figure 4 jimaging-08-00255-f004:**
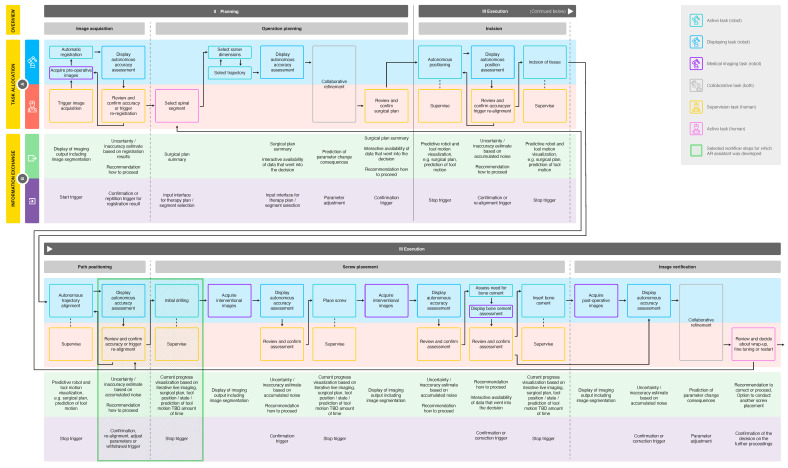
In depth workflow analysis of a futuristic PSP procedure, including task allocation (**A**) and information exchange (**B**) between surgeon and robotic system. Allocated tasks were categorized, and two workflow steps were selected for AR-assistant development (see legend).

**Figure 5 jimaging-08-00255-f005:**
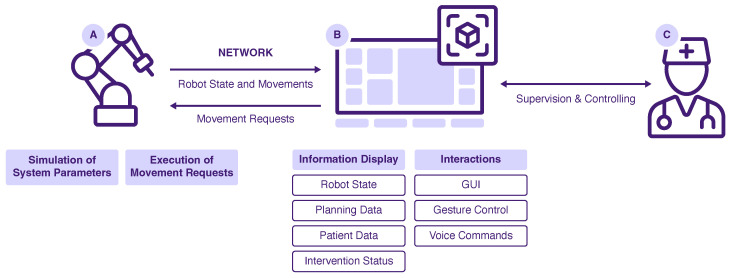
System overview of the developed prototype: Robot (**A**), AR System (**B**) and User (**C**).

**Figure 6 jimaging-08-00255-f006:**
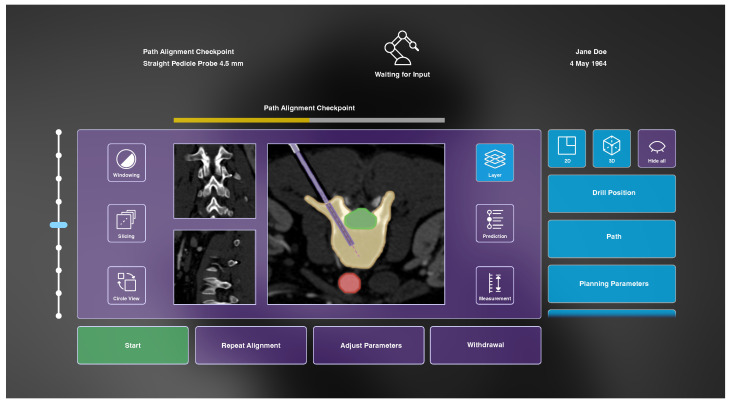
The developed AR interface during the *path alignment checkpoint* before *initial vertebra drilling* including a heads-up status bar (**top**) displaying general information and a dynamic GUI to enable system–user interaction (**bottom**).

**Figure 7 jimaging-08-00255-f007:**
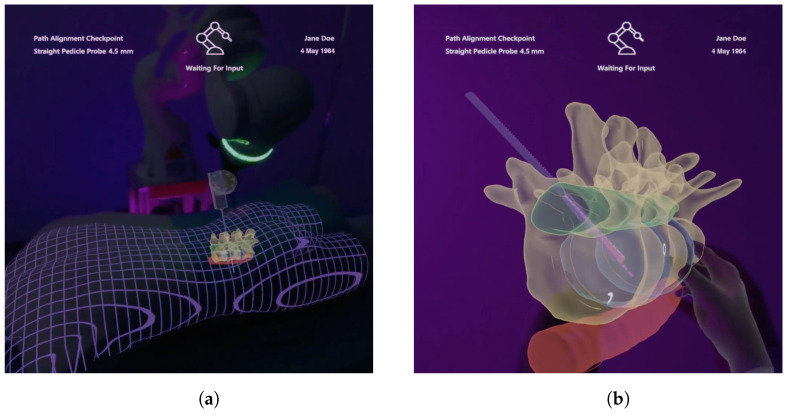
Included 3D models under AR view which contain information about the anatomy of interest, surrounding anatomy, the drill and planning parameters. (**a**) Registered visualization. (**b**) Enlarged virtual copy.

## Data Availability

Not applicable.

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
