# Peer review of "AR-Supported Supervision of Conditional Autonomous Robots: Considerations for Pedicle Screw Placement in the Future"

_2313-433X, 2022, doi:10.3390/jimaging8100255_

Round 1
Reviewer 1 Report
1. This is an interesting manuscript to outline future scenarios for robotic PSPs. In the "abstract", the author needs to explain some details of the AR prototype, especially some functional parameters of the user interface.
2. In the “Introduction”, the authors have mentioned open procedures and minimally invasive screw placement. How did the techniques used in this manuscript affect them?
3. In the 1.2 Related Work, the content of this sentence is not clear: “Our project tackles a research field of the mid-term future. Still, current research covers some aspects we already included into our work.” The meanings of “a research field “or “some aspects”? should be more clear.
4. Although the authors mention it in the introduction, the authors should explain the advantages or differences between their software and the software found in FDA-approved commercial systems.
5. Why the usage of AR for robot-assisted surgeries is low? Maybe it has some relation with this manuscript.
6. In order to facilitate the reader to refer to the design, the parameters of these hardware components of the new system in 3.3.2 can be given more in the manuscript.
Reviewer 2 Report
The presented manuscript presents future scenarios for roboter assisted and robotic pedicle screw placement (PSP) under supervision of a surgeon supported by augmented reality (AR).
The whole manuscript is pretty sound, but however there are some aspects to include and further discuss:
PSP is not only performed in orthopedia but also in trauma- and neurosurgery. Especially in neurosurgery the concept of navigation and AR is rouitnely used and has also been applied to spine surgery in some centers in recent years, even though still not widespread used. Does this have any effect on the future scenarios?
The authors could underpin the “need” for robot assistance by recent data on free-hand vs. navigated vs. robot assisted screw placement. How does the surgical approach (open vs. MIS) affect the future scenario?
As accurate patient registration and accurate transformation of preoperative data onto intraoperative data is a prerequisite for navigated PSP, how can this be achieved, and what are the prerequisite for the future scenarios under this aspect?
How does the presented “system” differ from nowadays available systems, what are the advantages or challenges?
